# Clinical Support through Telemedicine in Heart Failure Outpatients during the COVID-19 Pandemic Period: Results of a 12-Months Follow Up

**DOI:** 10.3390/jcm11102790

**Published:** 2022-05-16

**Authors:** Paolo Severino, Andrea D’Amato, Silvia Prosperi, Michele Magnocavallo, Annalisa Maraone, Claudia Notari, Ilaria Papisca, Massimo Mancone, Francesco Fedele

**Affiliations:** 1Department of Clinical, Internal, Anesthesiology and Cardiovascular Sciences, Sapienza University of Rome, Viale del Policlinico 155, 00161 Rome, Italy; damatoandrea92@gmail.com (A.D.); silviapro@outlook.it (S.P.); michelefg91@gmail.com (M.M.); claudia.notari2@gmail.com (C.N.); ilaria.papisca@gmail.com (I.P.); massimo.mancone@uniroma1.it (M.M.); francesco.fedele@uniroma1.it (F.F.); 2Department of Human Neurosciences, Sapienza University of Rome, 00161 Rome, Italy; annalisa.maraone@uniroma1.it

**Keywords:** heart failure, telemedicine, mortality, hospitalization, MACE, COVID-19

## Abstract

Background: Heart failure (HF) patients are predisposed to recurrences and disease destabilizations, especially during the COVID-19 outbreak period. In this scenario, telemedicine could be a proper way to ensure continuous care. The purpose of the study was to compare two modalities of HF outpatients’ follow up, the traditional in-person visits and telephone consultations, during the COVID-19 pandemic period in Italy. Methods: We conducted an observational study on consecutive HF outpatients. The follow up period was 12 months, starting from the beginning of the COVID-19 Italy lockdown. According to the follow up modality, and after the propensity matching score, patients were divided into two groups: those in G1 (*n* = 92) were managed with traditional in-person visits and those in G2 (*n* = 92) were managed with telephone consultation. Major adverse cardiovascular events (MACE) were the primary endpoints. Secondary endpoints were overall mortality, cardiovascular death, cardiovascular hospitalization, and hospitalization due to HF. Results: No significant differences between G1 and G2 have been observed regarding MACE (*p* = 0.65), cardiovascular death (*p* = 0.39), overall mortality (*p* = 0.85), hospitalization due to acute HF (*p* = 0.07), and cardiovascular hospitalization (*p* = 0.4). Survival analysis performed by the Kaplan–Meier method also did not show significant differences between G1 and G2. Conclusions: Telephone consultations represented a valid option to manage HF outpatients during COVID-19 pandemic, comparable to traditional in-person visits.

## 1. Introduction

Heart failure (HF) is a multisystemic disease characterized by repeated hospitalizations and progressive worsening. The correct follow up and therapy titration, particularly during the vulnerable phases among hospitalizations, remain a main target to improve patients’ prognosis [1,2,3,4]. The 2021 ESC Guidelines for the diagnosis and treatment of acute and chronic heart failure recommend that patients, after HF hospitalization, should undergo post discharge outpatient clinical visits following a precise timing [1]. However, studies highlight that missed outpatient clinical appointments are common [5,6]. This fact could be partially explained by the difficulty of reaching the medical examination due to health or logistic problems, such as forgetting appointments or administrative errors [7]. Moreover, the large number of HF patients causes a significant prolongation of the waiting time for HF outpatients’ visits, making a follow up according to the times defined by Guidelines impractical [1].

During 2019 and 2020, hospital admissions for HF were drastically reduced due to the Coronavirus Disease 2019 (COVID-19) outbreak [8,9,10]. The COVID-19 related pandemic outbreak forced the adoption of severe restraint measures worldwide, such as social isolation [11,12]. The COVID-19 outbreak and the related lockdown had a great impact, from both a medical and psychological point of view, on HF patients [13,14]. Several hospitals and their specialist teams were converted to specific COVID-19 healthcare centers, and, therefore, outpatients’ services and regular follow up visits were drastically reduced [15].

HF patients are a commonly known frail population that are predisposed to recurrences and disease destabilizations with worse outcomes, and these have occurred even more frequently during the COVID-19 outbreak period [16,17,18]. Cardiovascular deaths increased by 8%, with a 23% rise in HF related deaths [19]. However, the number of patients’ medical examinations has declined, likely due to fear and social restrictions, and a paradoxical reduction in HF hospitalization rates was observed [6,7,8]. In Italy, a decrease of 49% in acute HF hospital admissions from February to April 2020 was recorded compared with the same period in 2019 [20]. A similar trend has also been reported in Denmark, Germany, and the USA [10,21,22].

In this scenario, telemedicine emerged as a proper way to ensure continuous care using smartphones apps and telephone or video calls. Although the HF Guidelines stress the importance of telemedicine [1] and many studies demonstrate that it is associated with a reduction in all-cause mortality and HF hospitalization [23,24,25], its use does not belong to the routine approach yet, showing a low class of evidence [1]. The COVID-19 pandemic has completely changed physicians’ approach to telemedicine, highlighting its potential advantages [24,25].

The purpose of this study was to compare virtual management, performed through telephone consultation, with traditional in-person visits in a population of HF outpatients in terms of major adverse cardiovascular events (MACE) and overall mortality, cardiovascular death, hospitalization due to acute HF and cardiovascular hospitalization at 12 months follow up.

## 2. Methods

We conducted an observational study, enrolling consecutive HF outpatients previously hospitalized at the Department of Clinical, Internal, Anesthesiology, and Cardiovascular Sciences of Sapienza University of Rome, and all were managed by our outpatients HF follow up service for at least one year. The inclusion criteria were patients with HF diagnosis according to the Guidelines [1] who were hospitalized at the Cardiovascular Sciences Department of Sapienza University of Rome and are over the age of 18. The exclusion criteria were current COVID-19 infection, a history or new diagnosis of major depressive disorder, a diagnosis of any malignancy reducing short-term life expectancy, and patients followed up with invasive telemonitoring systems, such as CardioMEMS™ and HeartLogic™. The follow up period duration was 12 months, starting from the beginning of the COVID-19 Italy lockdown on 9th March 2020.

According to the follow up modality, the patients were divided in two groups:Group 1 (G1): patients managed through traditional in-person visits.Group 2 (G2): patients managed through telephone consultations.

Traditional visits included patient interviews assessing New York Heart Association (NYHA) class and the measurement of blood pressure, body weight, and peripheral oxygen saturation. In addition, a 12-lead electrocardiogram and a complete objective cardiological examination were performed. If necessary, patients were subjected to an echocardiographic examination. Home therapy was evaluated and managed, if needed. Medical therapy was up titrated until the highest dose tolerated and optimized according to symptoms and hemodynamic status. In-person visits were performed at 6 and 12 months from the follow up beginning.

Virtual visits were conducted by cardiologists through telephone consultation, assessing changes in the clinical parameters. In particular, variations in blood pressure, body weight, and NYHA class were assessed. Telephone consultations were performed at 3, 6, and 12 months from the follow up beginning.

A Meta-analysis Global Group in Chronic Heart Failure (MAGGIC) score has been assessed for each patient of the two groups using the web application at http://www.heartfailurerisk.org/ (accessed on 9 March 2020).

The primary outcome of the study was to evaluate the incidence of MACE by means of a composite of overall mortality, acute myocardial infarction, stroke, and hospitalization due to HF at 12 months. The single secondary outcomes were overall mortality, cardiovascular death, hospitalization due to HF, and cardiovascular hospitalization.

Propensity score matching analysis was used to homogenize the numerical differences between the two populations. Figure 1 reports the study design.

Baseline epidemiological, clinical, and echocardiographic parameters were retrospectively collected by checking the clinical records for each patient.

All data records were collected in a dedicated Excel Database. The study was conducted according to the Helsinki Declaration. The study protocol was approved by the Ethical Committee of Policlinico Umberto I of Rome.

### Statistical Analysis

Propensity score matching was performed to reduce the risk of selection bias. Patients were divided into two cohorts: patients managed through traditional in-person visits (G1) and patients managed through telephone consultations (G2). Due to differences in the key baseline characteristics, we used propensity score matching for the two cohorts and assembled a cohort for each comparison; all of the measured covariates were well-balanced across the comparator groups. The propensity score is defined as the subject’s probability of receiving a specific treatment or exposure (in this case, telephone consultation) given a set of measured baseline covariates. A logistic regression model was used to obtain propensity scores, with the telephone consultation protocol defined as the dependent variable, and age, gender, clinical characteristics, and echocardiographic parameters entered as covariates. Matching was performed using the nearest neighbor matching protocol (matching ratio of 1 to 1 without replacement) and a caliper width of 0.01. The balance of characteristics was assessed by estimating the standardized differences between groups; standardized difference indicates the degree of systematic differences in the covariates between groups. Operationally, a standardized difference >10% represents a meaningful imbalance in a given variable between groups.

The normal distribution of variables was assessed with the Kolmogorov–Smirnov test. For continuous variables, descriptive statistics were provided (number of available observations, mean, standard deviation), while the median (interquartile range) was used for non-normal data. Categorical data were described as the number (percentage). Student’s *t*-test, the χ^2^ test, and the Fisher exact test were used for comparisons. For all tests, a *p*-value < 0.05 was considered statistically significant.

The Kaplan–Meier method was used to estimate the cumulative event rates in the two groups. Differences in each group were compared using log-rank tests. The Cox regression hazard model was performed to obtain the hazard ratio (HR) for the endpoints.

The annual rate of death was estimated based on the MAGGIC scores and compared with the annualized observed rate of death aiming at calculating the % risk reduction.

All statistical analyses were performed using STATA statistical analysis software (version 16) (StataCorp. 2019. Stata 16 Base Reference Manual. Stata Press, College Station, TX, USA).

## 3. Results

A total of 1716 consecutive patients were included in the study. All patients were referred to the HF-dedicated outpatients service of Policlinico Umberto I of Rome. During a period of 12 months, starting from the beginning of the COVID-19 lockdown in Italy on 9th March 2020, 1624 patients continued the traditional follow up (G1), while 92 patients were managed with telephone consultation (G2). The follow up mean time was 11.5 ± 1.7 months for G1 patients and 11.3 ± 2.1 months for G2 patients. The baseline characteristics of the overall population are shown in Table 1. Considering the overall population, 1149 (67%) patients were female. The mean age was 70.5 ± 12.9. Arterial Hypertension was present in 1380 (80.4%) patients, diabetes mellitus in 283 (16.5%) patients, dyslipidemia in 935 (54.5%) patients, smoking habits in 525 (30.6%) patients, and familiarity for cardiovascular diseases in 641 (37.4%) patients. The average left ventricular ejection fraction (LVEF) was 42 ± 12% (Table 1).

Statistically significant differences between G1 and G2 were found regarding female gender (*p* = 0.001), diabetes mellitus (*p* = 0.001), smoking habits (*p* = 0.001), familiarity for cardiovascular diseases (*p* = 0.01), and average LVEF (*p* = 0.001) (Table 1). No differences have been found between the two groups regarding the MAGGIC score (*p* = 0.72).

Subsequently, we performed a propensity matching score analysis to compare the two groups. The baseline, echocardiographic, and therapy features of the two groups after propensity matching score are shown in Table 2.

At the end of the 12-months follow up, therapy was modified in 15 patients of G1. In particular, five patients increased the loop diuretic dose, while three patients decreased it. Six patients increased the dose of angiotensin-converting enzyme inhibitors/angiotensin receptor blockers (ACEi/ARBs), while two patients decreased it. Two patients switched from ACEi/ARBs to angiotensin receptor-neprilysin inhibitor (ARNI). Two patients increased the dose of beta blockers (BB). The loop diuretic suspension occurred in one patient.

At the end of the 12-months follow up, the therapy was modified in 19 patients of G2. In particular, four patients increased the dose of loop diuretic, while four patients decreased it. Five patients increased the ACEi/ARBs dose, while three patients decreased it. One patient switched from ACEi/ARBs to ARNI. Three patients increased the dose of BB. No loop diuretic suspension occurred.

Adverse events occurrence in the two groups during the follow up period is shown in Table 3. At the end of the follow up, no significant differences in the defined endpoints were observed. In particular, no differences between G1 and G2 have been observed regarding the primary endpoint MACE (20 vs. 18; HR 1.15; 95% CI 0.6–2.19; *p* = 0.65), as well as the single secondary endpoints of cardiovascular death (7 vs. 4; HR:1.72; 95% CI 0.5–5.89; *p* = 0.39), overall mortality (11 vs. 10; HR: 1.09; 95% CI 0.46–2.56; *p* = 0.85), hospitalization due to HF (18 vs. 9; HR: 2.07; 95% CI 0.93–4.61; *p* = 0.07), and cardiovascular hospitalization (19 vs. 14; HR 1.34; 95% CI 0.67–2.68; *p* = 0.4) (Table 3).

After, survival analysis was performed through the Kaplan–Meier method, no differences between the two groups were observed regarding MACE (Figure 2) and each single secondary endpoint (Figure 3). Subsequently, we compared the expected and observed mortality, through the MAGGIC score, between the two groups (Figure 4).

## 4. Discussion

The applicability of telemedicine in the diagnosis and follow up of cardiovascular diseases is constantly increasing [1,26]. Several studies underline the advantages of telemedicine use in patents with HF in terms of clinical and psychological success [24,25,26,27]. Salzano et al. demonstrated that telemedicine services expressly set up during the COVID-19 outbreak reduced the composite outcome of HF hospitalization and death when compared to the previous year without telemedicine [24]. Sammour et al. compared telemedicine visits in 2020 with in-person visits in the previous two years, proving that mortality was similar for both at 30 and 90 days (0.8% vs. 0.7% and 2.9% vs. 2.4% respectively), without the need for intensive care or hospital attendance with telehealth visits [25]. Galinier et al. demonstrated that some subgroups of patients in the selected population managed with telemonitoring may benefit in terms of clinical outcomes [26]. Wells et al. demonstrated that telehealth sessions improved life quality in patients with advanced HF [27]. Telemonitoring through invasive devices represents a promising tool for the management and follow up of HF patients. Visco et al. reported how telemonitoring allowed for the evaluation of the usefulness of new HF therapeutic approaches, such as cardiac contractility modulation, and their most appropriate timing for implantation [28].

Our study fits in this continuum, demonstrating that telephone consultation represented a valid method, in comparison to in-person visits, to manage HF outpatients during the COVID-19 pandemic period. In particular, we demonstrated no differences in terms of MACE (*p* = 0.65), overall mortality (*p* = 0.85), cardiovascular death (*p* = 0.39), hospitalization due to HF (*p* = 0.07), and cardiovascular hospitalization (*p* = 0.4) between a group of patients managed with telephone consultations and another group managed with traditional in-person visits.

The main goal of telemedicine during COVID-19 is to avoid worsening for HF patients, monitor parameters, manage therapy, and give psychological support [29]. Telemonitoring is also an important tool for maintaining a common thread with patients, which is useful for the clinician to stay up to date and for the patient to always feel under control and satisfied [30,31]. As previous stated, HF patients are a fragile and complex population, and they particularly suffered, from a clinical and psychological point of view, during the stressful period of the COVID-19 pandemic [32]. In this context, virtual visits may represent an adequate method to follow up HF patients during this period [33]. Structured telephone support, defined as monitoring, self-care management, or both, delivered using telephone calls [34] may represent the most simple and affordable system for HF centers starting with telemedicine during COVID-19. In fact, telephone consultation has been validated as a correct approach for cardiologists to identify those HF patients who need urgent visits, manage therapy, execute further diagnostic exams, and provide psychological support [35]. Other considerations may be deduced considering the heart failure with preserved ejection fraction (HFpEF) population and telemedicine. A significant proportion of heart failure with reduced ejection fraction (HFrEF) patients have cardiovascular implantable electronic devices (CIED), compared to HFpEF patients, who are much more rarely invasively telemonitored. HFpEF patients are an aged population with high mortality and hospitalization rates, representing an ever-growing population in the HF spectrum [36,37]. For these reasons, telemedicine approaches through telephone consultations or other external devices may be particularly beneficial for HFpEF patients.

Telemedicine is not without limits. It requires technological equipment, such us telephones, blood pressure machines, tablets, and other devices, that is very expensive, and not all patients are able to afford it [34]. Furthermore, HF patients are usually old and not always able to use these devices without any support [34].

In conclusion, the ongoing COVID-19 pandemic is overshadowing other diseases of global relevance, such as cancer and cardiovascular diseases, particularly HF. Moreover, adequate care has become even more difficult due to the employment of economic and human sources to face the COVID-19 pandemic. The impossibility of guaranteeing adequate care, particularly in term of timing, makes the use of alternative methods necessary to overcome current problems. Other clinical telemedicine experiences outside the HF context have already shown optimal results [38,39]. Telemedicine is not only spreading as a new medical approach in cardiology contexts but is also demonstrating its reliability and effectiveness in other fields [40,41].

Many targets have been achieved and many more are still to be pursued regarding the virtual follow up of HF patients: (i) the possibility of following up with all patients discharged after a hospitalization, according with the times suggested by the Guidelines [1] and major clinical trials, in order to prevent disease exacerbation and HF progression; (ii) the possibility of stratifying HF patients, identifying patients at short-term risk of disease exacerbation who can benefit from a long-term virtual follow up and reserving in-person visits for high-risk patients; (iii) the possibility of titrating therapy quickly, as suggested by the Guidelines [1], and improving treatment adherence [42]; (iv) the possibility of prescribing further exams to complete a diagnostic path; (v) the possibility of relieving the congestion of hospital HF outpatients’ services, reducing waiting time. These aspects may bring benefits in terms of patients’ care, hospital organization, and healthcare costs [43].

However, the identification of patients who may benefit more from virtual visits and those who may benefit from more in-person visits remains a major issue. Moreover, while there are precise indications regarding follow up times for in-person visits, they are missing regarding virtual visits for HF outpatients.

## 5. Study Limitations

The present observational study has some limitations. Our study sample is small, and several limitations in patients’ enrollment were due to the difficulties in collecting clinical data during the COVID-19 pandemic. The retrospective collection of baseline information from electronic health records does not warrant the standardization of data collection. Extensive data collection for each patient was not possible, especially during the initial phase of the pandemic. Several objective parameters were missed because signs and symptoms were investigated by telephone and not by checking the patients in-person.

## 6. Conclusions

Telephone consultation represented a valid option to manage HF outpatients who could not attend an HF outpatient in-person service during the COVID-19 pandemic outbreak period. Telephone consultation allows healthcare workers to maintain a contact with patients, evaluate if they are facing disease exacerbation, and, eventually, optimize their therapy with an effectiveness comparable to traditional in-person visits in terms of prognosis. In the future, beyond the pandemic period, telemedicine may be useful for HF patients’ follow up, reducing possible complications due to hospital attendance, such as infections, and waiting times for in-person visits. Although telemedicine should be encouraged, there are some unsolved issues. In particular, the correct identification of patients to be allocated to a virtual follow up and those that need in-person visits, as well as the correct timing of the follow up for virtual visits, must be accurately defined in the future.

## Figures and Tables

**Figure 1 jcm-11-02790-f001:**
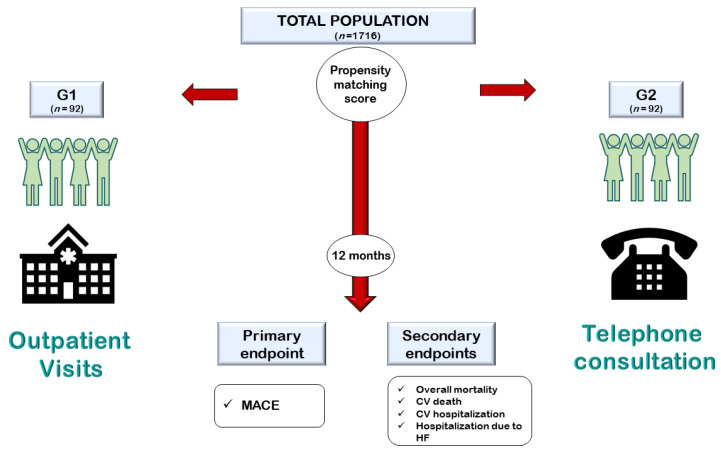
Representation of the study design. G1: group 1; G2: group 2; MACE: major adverse cardiovascular events; CV: cardiovascular; HF: heart failure.

**Figure 2 jcm-11-02790-f002:**
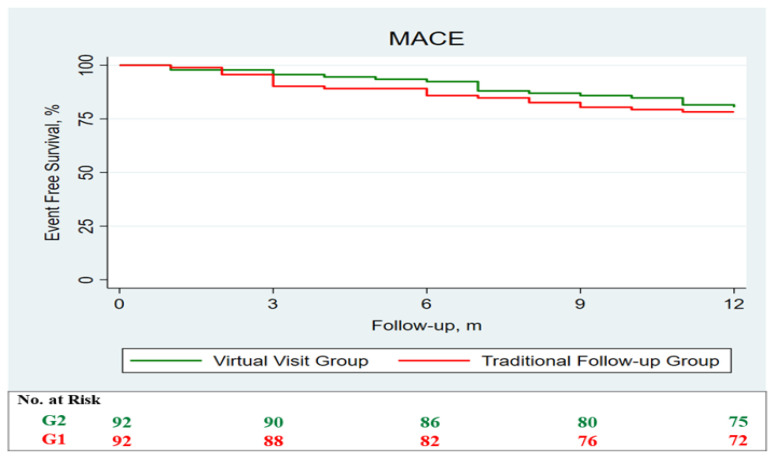
Survival analysis regarding the primary endpoint performed through the Kaplan–Meier method. Survival analysis demonstrates no differences in term of MACE, which is a composite of overall mortality, acute myocardial infarction, stroke, and hospitalization due to heart failure, between the two groups. G1 (red) is the group followed up through traditional in-person visits, while G2 (green) represents the virtual visit group.

**Figure 3 jcm-11-02790-f003:**
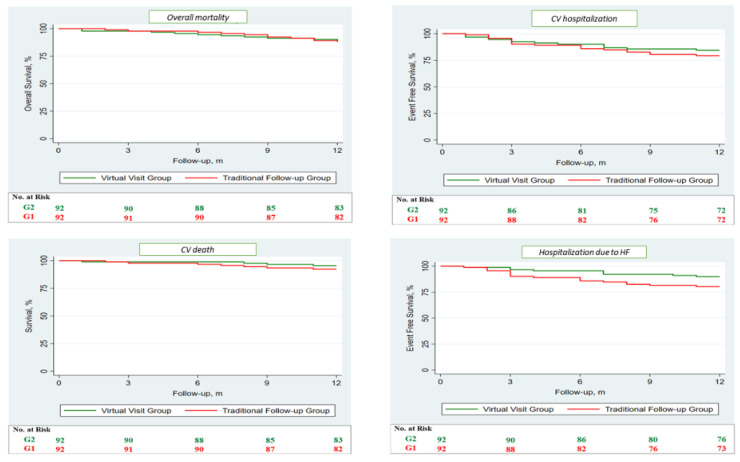
Survival analysis regarding the single secondary endpoints performed through Kaplan–Meier method. Survival analysis demonstrates no differences regarding the single secondary endpoints of overall mortality, cardiovascular death, cardiovascular hospitalization, and hospitalization due to HF between the two groups. G1 (red) is the group followed up through traditional in-person visits, while G2 (green) represents the virtual visit group. CV: cardiovascular; HF: heart failure.

**Figure 4 jcm-11-02790-f004:**
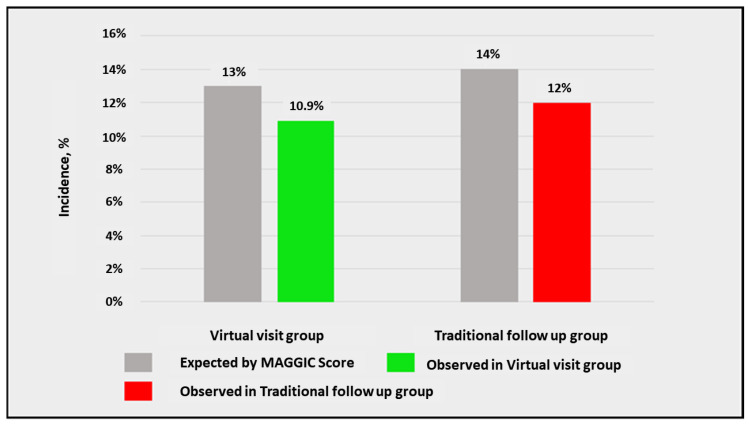
Comparison between expected and observed mortality, assessed through the Meta-analysis Global Group in Chronic Heart Failure (MAGGIC) score, between the two groups.

**Table 1 jcm-11-02790-t001:** Overall baseline characteristics.

	Overall Population *n* = 1716	G1 *n* = 1624	G2 *n* = 92	*p*-Value
Age (±SD)	70.5 ± 12.9	70.4 ± 13	72.6 ± 11.5	0.11
Female Gender, *n* (%)	1149 (67)	1118 (68.8)	31 (33.7)	<0.001
Arterial Hypertension, *n* (%)	1380 (80.4)	1308 (80.5)	72 (78.3)	0.59
Diabetes mellitus, *n* (%)	283 (16.5)	255 (15.7)	28 (30.4)	<0.001
Dyslipidemia, *n* (%)	935 (54.5)	876 (53.9)	59 (64.1)	0.05
Smoking habit, *n* (%)	525 (30.6)	472 (29.1)	53 (57.6)	<0.001
Familiarity for CVD, *n* (%)	641 (37.4)	594 (36.6)	47 (51.1)	0.01
Creatinine Clearance, mL/min (±SD)	68.9 ± 25.2	66.8 ± 25.4	68.5 ± 22.5	0.52
LVEF, % (±SD)	42 ± 12	42 ± 12	48 ± 10	<0.001
MAGGIC Score	21 ± 7	21 ± 7	20 ± 7.8	0.44

CVD: cardiovascular diseases; HF: heart failure; LVEF: left ventricular ejection fraction; MAGGIC: meta-analysis global group in chronic heart failure.

**Table 2 jcm-11-02790-t002:** Baseline characteristics after propensity matching score analysis.

	G1 *n* = 92	G2 *n* = 92	*p*-Value
Age (±SD)	71.2 ± 13	72.6 ± 11.5	0.42
Female Gender, *n* (%)	27 (29.3)	31 (33.7)	0.53
Arterial hypertension, *n* (%)	73 (79.3)	72 (78.3)	0.85
Diabetes mellitus, *n* (%)	28 (30.4)	28 (30.4)	1
Dyslipidemia, *n* (%)	65 (70.7)	59 (64.1)	0.35
Smoking habit, *n* (%)	52 (56.5)	53 (57.6)	0.89
Familiarity for CVD, *n* (%)	47 (51.1)	47 (51.1)	1
Creatinine Clearance, mL/min (±SD)	66.6 ± 25.1	68.5 ± 22.5	0.60
Ischemic HF etiology, *n* (%)	46 (50)	54 (59)	0.30
HFpEF, *n* (%)	43 (47)	48 (52)	0.55
HFmrEF, *n* (%)	16 (17)	15 (16)	1
HFrEF, *n* (%)	33 (36)	29 (32)	0.64
LV EDD, mm (±SD)	53.3 ± 7.6	54 ± 6.6	0.5
IVS, mm (±SD)	11.35 ± 1.75	11.54 ± 1.72	0.46
PW, mm (±SD)	10.2 ±1.5	10 ± 1.36	0.34
LVEF, % (±SD)	47 ± 11	48 ± 10	0.76
TAPSE, mm (±SD)	18 ± 4	17 ± 4	0.09
PAPs, mmHg (±SD)	40 ± 11	37.5 ± 13	0.16
E/e’ ratio (±SD)	9 ± 2	9.5 ± 2.2	0.11
BB, *n* (%)	80 (87)	74 (81)	0.32
ACE-i/ARBs, *n* (%)	63 (68)	66 (72)	0.75
ARNI, *n* (%)	23 (25)	13 (14)	0.09
MRAs, *n* (%)	47 (51)	52 (57)	0.55
Loop diuretics, *n* (%)	64 (70)	51 (55)	0.07
MAGGIC Score	20 ± 7.2	20 ± 7.8	0.72

CVD: cardiovascular diseases; HF: heart failure; HFpEF: heart failure with preserved ejection fraction; HFmrEF: heart failure with mildly reduced ejection fraction; HFrEF: heart failure with reduced ejection fraction; LV EDD: left ventricular end-diastolic diameter; IVS: interventricular septum; PW: posterior wall; LVEF: left ventricular ejection fraction; TAPSE: tricuspid annular plane systolic excursion; PAPs: pulmonary artery systolic pressure; BB: betablockers; ACE-i/ARBs: angiotensin-converting enzyme inhibitors/angiotensin receptor blockers; ARNI: angiotensin receptor-neprilysin inhibitor; MRAs: mineralocorticoid receptor antagonists; MAGGIC: meta-analysis global group in chronic heart failure.

**Table 3 jcm-11-02790-t003:** Adverse cardiovascular events during the follow up period.

Outcome	G1 *n* = 92	G2 *n* = 92	HR	95% CI	*p*-Value
MACE, *n* (%)	20 (21.7)	18 (19.7)	1.15	(0.61–2.19)	0.65
Overall mortality, *n* (%)	11 (12)	10 (10.9)	1.09	(0.46–2.56)	0.85
CV death, *n (%)*	7 (7.3)	4 (4.3)	1.72	(0.50–5.89)	0.39
Stroke/TIA, *n* (%)	1 (1.1)	1 (1.1)	0.99	(0.06–15.90)	1
AMI, *n* (%)	2 (2.2)	3 (3.3)	0.66	(0.11–4.02)	0.66
CV Hospitalization, *n* (%)	19 (20.7)	14 (15.2)	1.34	(0.67–2.68)	0.40
Hospitalization due to HF, *n* (%)	18 (19.6)	9 (9.8)	2.07	(0.93–4.61)	0.07
Follow up, months (±SD)	11.5 ± 1.7	11.3 ± 2.1			0.65

MACE: major adverse cardiovascular events; CV: cardiovascular; TIA: transient ischemic attack; AMI: acute myocardial infraction; HF: heart failure; HR: hazard ratio; CI: confidence interval.

## Data Availability

The data presented in this study are available on request from the corresponding author.

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
