# Peer review of "Clinical Support through Telemedicine in Heart Failure Outpatients during the COVID-19 Pandemic Period: Results of a 12-Months Follow Up"

_jcm, 2022, doi:10.3390/jcm11102790_

Round 1
Reviewer 1 Report
The manuscript by Paolo Severino et al. entitled “Clinical support through telemedicine in heart failure outpatients during COVID-19 pandemic period: results of a 12-months follow up” aimed to compare virtual management, performed through telephone consultation, with traditional in-person visits, in a population of HF outpatients in term of major adverse cardiovascular events (MACE), and overall mortality, cardiovascular death, hospitalization due to acute HF and CV hospitalization, at 12 months follow-up.
The article is well written and leads some evidence to such point; however, some major issues need to be addressed to improve the significance and reliability of the results of the study:
-Firstly, the exclusion and inclusion criteria are not clear. For example, it is important to know if patients with telemonitoring devices, such as Cardiomems or Heart logic system, were excluded. Please, clarify this aspect in the Methods.
-It would be important to report in Table 2 the different classes of drugs taken by patients. Were the patients all taking optimal medical therapy? Was the rate of therapy changes the same in the two groups?
-It would be important to report echocardiographic parameters of right heart function, such as TAPSE. Please, add these parameters in Table 2.
-During the last years, the management of HF made substantial progress, focusing on device-based therapies to meet the demands of this complex syndrome. Specifically, in the management of this syndrome, therapeutic strategies usually aim for improved outcomes in terms of reduced mortality and fewer unplanned HF hospitalizations. Accordingly, the CHAMPION trial clearly showed a benefit of the pressure-guided therapy in terms of reduced hospitalizations for HF exacerbations. Consequently, authors should mention the role of telemonitoring devices in HF patients. These articles could be cited: doi: 10.3390/medicina58020210, DOI:10.3389/fcvm.2022.874433, DOI: 10.2174/0929867328666201218122633
As a result, the Reviewer suggests reconsidering the article after major revision.
Author Response
1-Firstly, the exclusion and inclusion criteria are not clear. For example, it is important to know if patients with telemonitoring devices, such as Cardiomems or Heart logic system, were excluded. Please, clarify this aspect in the Methods.
- We thank the reviewer for the comment. We added a clarification in the text about inclusion and exclusion criteria. Specifically, patients with cardiomems and invasive telemonitoring devices were excluded from the group 2.
2-It would be important to report in Table 2 the different classes of drugs taken by patients. Were the patients all taking optimal medical therapy? Was the rate of therapy changes the same in the two groups?
- We thank the reviewer for the comment. We added data about classes of drugs in Table 1 and 2. Therapy was optimized according to previous guidelines because the study was carried out in 2020; moreover, therapy was optimized according to patients' tolerance and hemodynamic status. We also added in the section “results” data about therapy changes in the two groups.
3-It would be important to report echocardiographic parameters of right heart function, such as TAPSE. Please, add these parameters in Table 2.
- We thank the reviewer for the comment. We added echocardiographic parameters in Table 1 and 2.
4-During the last years, the management of HF made substantial progress, focusing on device-based therapies to meet the demands of this complex syndrome. Specifically, in the management of this syndrome, therapeutic strategies usually aim for improved outcomes in terms of reduced mortality and fewer unplanned HF hospitalizations. Accordingly, the CHAMPION trial clearly showed a benefit of the pressure-guided therapy in terms of reduced hospitalizations for HF exacerbations. Consequently, authors should mention the role of telemonitoring devices in HF patients. These articles could be cited: doi: 10.3390/medicina58020210, DOI:10.3389/fcvm.2022.874433, DOI: 10.2174/0929867328666201218122633
- We thank the reviewer for the comment. We added these refences in text, mentioning the role of telemonitoring devices in HF patients.
Reviewer 2 Report
This study compared the prognosis of outpatients with heart failure during the COVID19 pandemic who were followed as usual and those who were followed by telephone and concluded that there were no significant differences between the two groups. While the focus of this study is interesting and useful in clinical practice, I would like to highlight the following key concerns that need to be addressed to improve the quality of this article.
- About 5% of the entire cohort is Group 2, but it is unclear how patients were selected.
- Medications are considered an important prognostic factor in the treatment of heart failure, but have not been studied.
- The etiology of heart failure, e.g., ischemic heart disease or not, should also be considered.
- The proportion of HFpEF also needs additional consideration.
- The KM curve needs to include the number of patients.
Author Response
1-About 5% of the entire cohort is Group 2, but it is unclear how patients were selected.
- We thank the reviewer for the comment. This study is a pilot study born during the COVID-19 pandemic to face the problems derived from pandemic period, regarding the follow up of our HF outpatients. We tried to evaluate the effectiveness and feasibility of an alternative approach to manage HF outpatients after hospitalization, comparing it to the standard approach. The 92 patients were selected on the basis of their individual compliance to be telemonitored, their reliability and a possible presence of a caregiver.
2-Medications are considered an important prognostic factor in the treatment of heart failure but have not been studied.
- We thank the reviewer for the comment. We added data about medications in Table 1 and 2.
3-The etiology of heart failure, e.g., ischemic heart disease or not, should also be considered.
- We thank the reviewer for the comment. We added this data in Table 1 and 2.
4-The proportion of HFpEF also needs additional consideration.
- We thank the reviewer for the comment, we added data regarding HFpEF patients in Table 1 and 2. Moreover we added some considerations regarding HFpEF group also in the text.
5-The KM curve needs to include the number of patients.
- We thank the reviewer for the comment. We added the number of patients on KM curves.
Round 2
Reviewer 1 Report
The Reviewer thanks the authors for their comprehensive answers.
Reviewer 2 Report
This manuscript is well revised. There are no further comments. Thank you for your response.